# Reduced Sleep Amount and Increased Sleep Latency in Prisoners: A Pilot Study in an Italian Jail

**DOI:** 10.3390/brainsci13010132

**Published:** 2023-01-12

**Authors:** Giulia D’Aurizio, Beatrice Tosti, Daniela Tempesta, Lucia Avvantaggiato, Alessandra Splendiani, Simona Sacco, Laura Mandolesi, Giuseppe Curcio

**Affiliations:** 1Department of Biotechnological and Applied Clinical Sciences, University of L’Aquila, 67100 L’Aquila, Italy; 2Department of Penitentiary Administration, Ministry of Justice, 00164 Roma, Italy; 3Department of Humanities, University of Naples Federico II, 80138 Naples, Italy

**Keywords:** actigraphy, insomnia, health, depression, anxiety, well-being, inmates, PSQI

## Abstract

Several previous subjective- or interview-based reports indicated a reduced sleep quality and quantity as well as a great incidence of insomnia troubles in prisoners living in jail. The aim of the present study is to assess the quality and quantity of sleep by using, for the first time, actigraphy. A total of thirty male prisoners and thirty male control volunteers accepted to participate in this study: to this end, they filled in some questionnaires to assess state and trait anxiety, depression, sleep quality, and insomnia severity. In addition, their sleep was actigraphically recorded for seven consecutive nights. The main results indicate a worsened mood in prisoners than in controls (with increased anxiety and depression) as well as a subjectively reported low sleep quality (higher scores at PSQI) with a clinical presence of insomnia complaints (as indicated by ISI scores). Moreover, objectively assessed sleep by means of actigraphy exhibited some worrying results, namely a longer sleep onset and a reduced total sleep time was seen in prisoners with respect to controls. The results have been discussed in the light of potential effects of sleep quality and quantity as well as of mood symptoms on cognitive functioning, as well as with respect to prisoners’ health and well-being.

## 1. Introduction

Sleep is an active, repetitive, and reversible behavior hypothesized to serve several functions, occurring throughout the brain and the body [1]. Focusing on cognitive functioning and behavior regulation, it has been reported that sleep loss or poor sleep quality can negatively influence several brain functions such as executive function [2,3,4], emotional processing [5,6], learning and memory consolidation [7,8], different aspects of behavior [9,10,11], and well-being [12,13]. Moreover, when a prolonged sleep deprivation occurs or an extended period is characterized by a systematic impairment of sleep quality and quantity, some psychiatric disturbances such as depressed mood, aggressive behavior, and anxiety can be observed [8,9,10,14,15,16]. As a consequence, most research looked at these effects on cognitive functioning, psychiatric symptoms, and psychophysical well-being in healthy and typical populations; very few studies investigated such a complex relationship in a very specific and peculiar population such as people living in prison [17,18].

As suggested by Morris and coworkers [19], incarceration can disrupt healthy sleep, and insomnia is associated with psychiatric symptoms and poor general medical health in this population. Recently, considerable legal discussions have been arisen over sleep deprivation in all jails and prisons that took into consideration several possible conditions of incarceration able to disrupt sleep quality and quantity, such as noise, inadequate bedding, constant illumination, medication restrictions, or early wake-up times. The authors concluded with the need for policies aimed at reducing unnecessary sleep deprivation and promote healthy sleep within correctional environments.

On the other hand, several experimental and empirical studies have also been carried out. In these studies, it has been reported that in a particular environmental context such as the prison, changes in sleep habits [20] and potential sleep disturbance (i.e., insomnia) can progressively increase with time [21], but little is known about “which” potential factors could qualitatively and quantitatively modulate sleep and “how” psychological and psychiatric variables could impact sleep.

Typically, sleep disturbances consequent to sleep pattern alteration represent a behavioral response linked to the prolonged exposure to extreme, stressful situations [22] such as earthquakes [23] or combat [24]. In these extreme situations, anxiety also plays a pivotal role in the development of both sleep and behavior alterations [15,25]. According to this point of view, living in prison could be considered as a prolonged stressful situation that can trigger not only a dysregulation of sleep patterns, but can also bring out psychiatric symptoms such as anxiety and/or depression.

Under the point of view of sleep quality and quantity, a very recent review [26] highlighted different typologies of measurement of sleep quality, inconsistent parameters of standardized measures, and methodological heterogeneity in the literature. Nonetheless, despite a sharp increase in the quantity of studies and some improved quality, the findings were highly variable and usually not replicable.

In a very recent study [27], it has been shown that an increased anxiety state level and the presence of mood alteration corresponds to an increase in both poor sleep quality and, more specifically, insomnia complaints. The authors also showed that this effect could be modulated by the total amount of time spent in prison; this variable was described to have a stabilizing paradox function on anxiety and insomnia.

Essentially, all these studies, due to the restrictions of the prisons, have been carried out by using subjective measures or interview-based reports. To the best of our knowledge, there is a complete lack of objective assessment of sleep through reliable and well-validated devices able to detect and measure sleep quality, quantity and, hopefully, architecture. Actigraphy is a noninvasive, cheap, and safe device able to provide answers to this kind of questions related to sleep medicine in particular environments such as prisons. Additionally, this could strongly contribute to improve the quality of life and well-being of the inmates. Moreover, the use of this assessment tool could help in planning and conducting studies aimed at improving sleep; by knowing sleep troubles and the altered sleep architecture, possible interventions (i.e., based on cognitive and behavioral therapies) could be initiated on this population.

The main scope of the present cross-sectional study is to assess, for the first time in Italian prisoners, the quality and quantity of actigraphically assessed sleep. To this aim, we compared actigraphic sleep from thirty prisoners and thirty age-matched volunteers. Our hypothesis was that prisoners’ sleep was quantitatively worse than in controls. Moreover, we were also interested in assessing anxiety, depression, sleep quality (through PSQI), and insomnia severity as well as their correlation. Our hypothesis, based on the existing literature, was that prisoners presented more mood troubles as well as a worsened quality of sleep and greater insomnia complaints.

## 2. Materials and Methods

### 2.1. Participants

Sixty male subjects were enrolled to participate in this study. The sample included prisoners and control groups. After obtaining the needed authorization by the penitentiary administration (Provveditorato Regionale Amministrazione Penitenziaria LAM-Lazio, Abruzzo, Molise), in collaboration with the internal psychological team of Penitentiary Institution of Lanciano (Chieti), we enrolled subjects asking them to participate in the study. Thirty male prisoners (Group 1; mean age 36.5 ± SD 9.0) accepted to participate. Each inmate voluntarily accepted to participate in the study, no incentives to participate were provided, and they were explicitly informed to feel free to leave the study at any time without any kind of repercussion. The control group consisted of 30 male volunteers (Group 2; mean age 35.5 ± SD 8.01). All participants were native Italian speakers with an education ranging between 5 and 8 years. This level of education assured that they were all able to read and understand the Italian version of the used questionnaires. Based on questionnaires and clinical interview, we excluded participants that were not free of medication, with a known neurological condition and medical condition upon the assessment, and with a known history of psychiatric disorder. 

The entire investigation was approved by the Regional Institutional Review Board (Provveditorato Regionale Amministrazione Penitenziaria LAM, n. PE 0026156, 24 December 2018, released to the CA) and was conducted according to the principles established in the Declaration of Helsinki.

### 2.2. Materials

All participants were asked to fill in a set of questionnaires to assess psychological and behavioral profile, as well as sleep characteristics. In addition, actigraphic recordings were collected for seven consecutive nights. More specifically, questionnaires and devices used for the assessment of sleep and psychological status are described below. 

(1) Beck Depression Inventory II Edition (BDI-2; Cronbach’s alpha = 0.89) [28,29] is a 21 multiple-choice self-report inventory for assessing affective–somatic (AS) and cognitive (C) dimensions of depression severity. The total score ranges from 0 to 63; in particular, a score from 10 to 18 indicates a mild-to-moderate depression, a score from 19 to 29 indicates a moderate-to-severe depression, and a score higher than 30 is indicative of a severe depression level [30].

(2) State-Trait Anxiety Inventory (STAI Form Y1 and Y2 [31,32] is a 40 multiple-choice items self-report inventory, divided into two sub-tests each having 20 items (for both state- and trait-anxiety) aimed at assessing and quantifying anxiety disorder in adults. All items are rated on a 4-point Likert scale. The recommended clinical cut-off is >46. The alpha reliability coefficients for the Y1 and Y2 are 0.92 and 0.90, respectively.

(3) Pittsburgh Sleep Quality Index (PSQI; Cronbach’s alpha = 0.83) [33,34] is a self-rated questionnaire for assess the overall sleep quality of the past month. It provides information about seven different components (subjective sleep quality, C1; sleep latency, C2; sleep duration, C3; habitual sleep efficiency, C4; sleep disturbances, C5; use of sleep medications, C6; daytime dysfunction, C7) that taken together form a global score. Usually, a PSQI global score ≥5 is an indicator of clinically significant sleep pathological alteration in at least two components or of moderate difficulties in more than three components.

(4) Insomnia Severity Index (ISI; Cronbach’s alpha = 0.90) [35,36] is a seven-items questionnaire widely used to quantify insomnia by evaluating some sleep aspects (i.e., sleep onset, sleep maintenance, early morning awakenings, sleep dissatisfaction, interference of sleep alterations with daytime functioning, sleep problems reported by others, level of distress secondary by the sleep alterations), referred to the past two weeks. The ISI total score ranges between 0 and 28, with a higher score indicating greater insomnia severity. More specifically, scores between 0 and 7 indicate a not significant insomnia, scores between 8 and 14 indicate subthreshold insomnia, scores between 15 and 21 indicate insomnia of moderate severity, and scores higher than 22 indicate severe insomnia [37].

(5) Actigraphy. The accelerometry-based activity monitor (Actiwatch Spectrum Plus–Philips Respironics, Amsterdam, Netherlands) was used to obtain activity counts in both groups. From the actigraphic data, by using the dedicated software, it was possible to derive different outcomes such as Total Sleep Time (TST), Sleep Efficiency (SE%), Sleep Latency, Wake After Sleep Onset (WASO), number and duration of Movements during sleep, etc.

### 2.3. Procedure

Before taking part in the study, informed consent was obtained from all participants. They were recruited individually and each of them was free to leave the study at any time. After enrolling in the study, each subject of both groups filled in BDI-2, STAI Y1 and Y2, PSQI, and ISI scales. Then, the collection of actigraphic sleep started and continued for seven consecutive nights. The study started in July 2021 and finished in February 2022.

### 2.4. Statistical Analyses

Statistical comparisons between prisoner and control subject groups were carried out by means of the Student’s *t*-test regarding age.

In order to explore the relationships between different scores at BDI-2, STAI Y1 and Y2, PSQI, ISI, and actigraphic data, a Pearson’s correlation was run.

Finally, one-way analysis of variance (ANOVA) was performed to test the differences between the two groups (Group 1, Group 2) with respect to sleep measures and all psychological score-dependent variables. With respect to the sleep measures, in the PSQI we considered the mean of all seven subscales and global score, in the ISI we considered all three scales (severity, impact, satisfaction) and the total score, while in the actigraphy, we considered the mean of TST (Total Sleep Time), SE (Sleep Efficiency), and WASO (Wake After Sleep Onset).

Alpha level was fixed to ≤0.05 and all tests were two-tailed. All statistical analyses were performed using IBM SPSS Statistics for Macintosh, version 27.01 (IBM Corp., Armonk, NY, USA).

## 3. Results

The comparison between age of prisoners (36.5 ± 9 years) and control subjects’ group (35.5 ± 8.01 years) did not show a statistically significant difference (t58 = 0.48; *p* = 0.62).

### 3.1. Correlation Analysis

Pearson correlation analysis showed a significant positive correlation between BDI-2 affective–somatic scale and PSQI total score (r = 0.38; *p* = 0.002), between BDI-2 affective–somatic scale and ISI satisfaction (r = 0.26; *p* = 0.047), between BDI-2 affective–somatic scale and both STAI-Y1 (r = 0.52; *p* < 0.001) and STAI-Y2 (r = 0.46; *p* < 0.001), between BDI-2 total score and PSQI total score (r = 0.33; *p* = 0.01), between BDI-2 affective–somatic scale and satisfaction (r = 0.3; *p* = 0.02) and total score (r = 0.26; *p* = 0.48) ISI subscale, between BDI-2 affective–somatic scale and both STAI-Y1 (r = 0.46; *p* < 0.001) and STAI-Y2 (r = 0.44; *p* < 0.001), between STAI-Y1 and PSQI global score (r = 0.6; *p* < 0.001), between STAY-Y1 and severity r = 0.38; *p* = 0.002) impact (r = 0.31; *p* = 0.016), satisfaction (r = 0.47; *p* < 0.00), and ISI total scores (r = 0.44; *p* < 0.001), between STAI-Y1 and sleep latency (r = 0.41; *p* < 0.001) and TST (r = 0.3; *p* = 0.02) derived from actigraphy data. A positive correlation was also found between STAI-Y2 scale and PSQI global score (r = 0.62; *p* < 0.001), as well as severity (r = 0.47; *p* < 0.001), impact (r = 0.31; *p* = 0.015) satisfaction (r = 0.47; *p* < 0.00), ISI total score (r = 0.45; *p* < 0.001), and sleep latency (r = 03; *p* = 0.021).

Moreover, as expected, the statistical analysis indicated a significant positive correlation between PSQI and ISI total score (r = 0.53; *p* < 0.001).

Main Pearson correlation results are illustrated in Figure 1.

All results from correlation analyses are reported in Table 1.

### 3.2. ANOVA Analysis

The one-way ANOVA showed a significant group effect on BDI affective–somatic performance (F_1,58_ = 5.7; *p* = 0.02; ηp2 = 0.09) and both Y1 (F_1,58_ = 73.77; *p* < 0.001; ηp2 = 0.56) and Y2 STAI scores (F_1,58_ = 44.16; *p* < 0.001; ηp2 = 0.43), indicating a higher BDI-2 and STAI score in Group 1 (BDI-2 = 8.02 ± 6.01; STAI Y1 = 47.27 ± 11.06; STAI Y2 = 46.37 ± 6.44) than in Group 2 (BDI-2 = 5.17 ± 3.5; STAI Y1 = 27.67 ± 5.83; STAI Y2 =35.5 ± 6.22).

Moreover, the one-way ANOVA on PSQI showed a significant main effect for group (F_1,58_ = 30.78; *p* < 0.001; ηp2 = 0.35), indicating higher PSQI total scores (reflecting poorer sleep quality) of Group 1 (10 ± 3.78) compared to Group 2 (5.17 ± 2.91; see Figure 2a). In addition, the following clinical domains of sleep difficulties assessed by PSQI questionnaire showed significant differences between Group 1 and Group 2: habitual sleep efficiency (F_1,58_ = 7.68; *p* = 0.007; ηp2 = 0.12; 4.17 ± 6.98 and 0.6 ± 1, respectively), sleep disturbances (F_1,58_ = 11.97; *p* = 0.001; ηp2 = 0.17; 1.73 ± 0.64 and 1.2 ± 0.55, respectively), and use of sleep medication (F_1,58_ = 6.73; *p* = 0.12; ηp2 = 0.01; 0.67 ± 1.21 and 0.07 ± 0.36, respectively).

With respect to ISI, the one-way ANOVA revealed a significant main group effect on ISI total score (F_1,58_ = 8.6; *p* = 0.005; ηp2 = 0.13) and severity (F_1,58_ = 7.86; *p* = 0.007; ηp2 = 0.12) subscale, indicating a poorer sleep quality in Group 1 (10.7 ± 8.2; 3.83 ± 3.37, respectively) compared to Group 2 (6.1 ± 3.9; 2 ± 1.2, respectively). The effect on total score is depicted in panel b of Figure 2.

Finally, with respect to objectively assessed sleep, the one-way ANOVA showed a significant group effect on TST (F_1,58_ = 4.87; *p* = 0.031; ηp2 = 0.08; Figure 3a) and latency (F_1,58_ = 24.15; *p* < 0.001; ηp2 = 0.3; Figure 3b), indicating a longer sleep latency and shorter TST in Group 1 (respectively, 2.02 ± 0.33 min and 402 ± 68.43 min) than in control group (respectively, 1.13 ± 0.73 min and 445 ± 83.4 min).

No other main effects on other considered variables were statistically significant.

## 4. Discussion

To the best of our knowledge, this is the first study aiming at assessing sleep both qualitatively and objectively in a group of prisoners. Living in a jail is considered as a condition able to disrupt healthy sleep and/or to induce clinical insomnia, and both are associated with psychiatric symptoms and poor general medical health [19]. Here, we assessed subjective (PSQI and ISI scores) and objective (through actigraphy) sleep as well as psychological status (depression and anxiety) in a group of prisoners, comparing them with a gender and age-matched control group.

A scenario of expected correlations has been observed between mood (depression and state-trait anxiety), quality of sleep (ISI and PSQI), and objectively assessed sleep. Depression resulted positively correlated with both state and trait anxiety, with insomnia complaints and sleep quality worsening. State and trait anxiety positively correlated with worsened sleep quality and insomnia problems, as well as with actigraphic measures of sleep latency and total sleep amount. Additionally, confirming previous studies, a significant positive correlation was observed between increased insomnia and worsened sleep quality.

When trying to compare prisoners (Group 1) and controls (Group 2), higher depression and state-and-trait anxiety were observed in the inmates’ group. Similarly, prisoners exhibited a poorer sleep quality as measured by PSQI and as reflected by worsened habitual sleep efficiency, greater incidence of sleep disturbances, and higher use of sleep medication. A companion effect was observed on insomnia complaints, with prisoners complaining significantly more about insomnia difficulties.

Finally, with respect to objectively assessed sleep, prisoners exhibited a longer sleep latency and a shorter total sleep time than controls, confirming the sleep difficulties already indicated with subjective measures such as PSQI and ISI.

These data provide strong support to some previous studies, by adding some original suggestions regarding objective sleep.

As a first, here we confirm the presence of relevant mood disorders that show their effect on increased depressive and anxious symptoms [19,27] that could in turn also favor substance abuse [38]. These complex interactions between prison environment and mood changes can contribute to significantly undermine the well-being and health of inmates. These effects can, in turn and as potential long-term effects, worsen also other psychological functions such as cognitive abilities.

With respect to subjective sleep quality and sleep complaints, we supported the previous literature that indicated that a complex environment such as the prison can induce changes in sleep habits [20] and potential sleep disturbances such as insomnia can progressively increase with time spent jail [21]. Additionally, in our sample, prisoners indicated a poorer quality of sleep and a higher incidence of insomnia than controls.

Finally, the most important point of this contribution does regard the objectively assessed sleep through the actigraphy technique. This is the first study aiming to provide quantitative data in support of the previous literature regarding sleep impairment and sleep complaints in prisoners [39,40]. Here, we show that subjective reports from inmates can be the reflection of structural and quantitative changes in daily sleep. As a first, a longer latency of sleep onset has been reported: this can negatively influence the quality of sleep and its restorative power by diminishing its potential satisfaction. A longer latency can also reflect the difficulties to fall asleep in an environment in which there is no possibility to individually choose when to “go” to sleep and when to “wake-up” and can strongly interact with mood disorders and particularly with anxiety. The same increase in anxiety can also be responsible for the significant reduction in total sleep amount we observed. 

These structural changes in sleep are of fundamental importance in light of some aspects that are very frequent among prisoners such as aggressive behaviors and the rate of suicidal attempts [41]. It is presumable that an improvement in mood (i.e., reduction in anxiety and depression) and a concomitant promotion of correct sleep hygiene could positively influence the affective life of the prisoners, by reducing altered emotional responses (for example, aggressive behaviors) and suicidal episodes. In this context, future research is mandatory in order to identify possible intervention strategies for improving inmates’ quality of life.

The present investigation is a pilot study that, for the first time, brings within the jail walls an objective technique to assess sleep, and thus it has several limitations. Small sample size, gender bias, and the impossibility to obtain more prolonged recording periods are just some of the main limits of this study. Moreover, we also have to take into account that this is a cross-sectional investigation and a pilot study; these aspects undermine its external validity and thus generalizability. Nonetheless, such limitations strongly depend on the objective experimental difficulties encountered during the study and are a direct consequence of the severe rules regulating the prison environment. Future directions may be carried out to reach more participants, in different conditions and jails, in order to try to overcome these limits.

## Figures and Tables

**Figure 1 brainsci-13-00132-f001:**
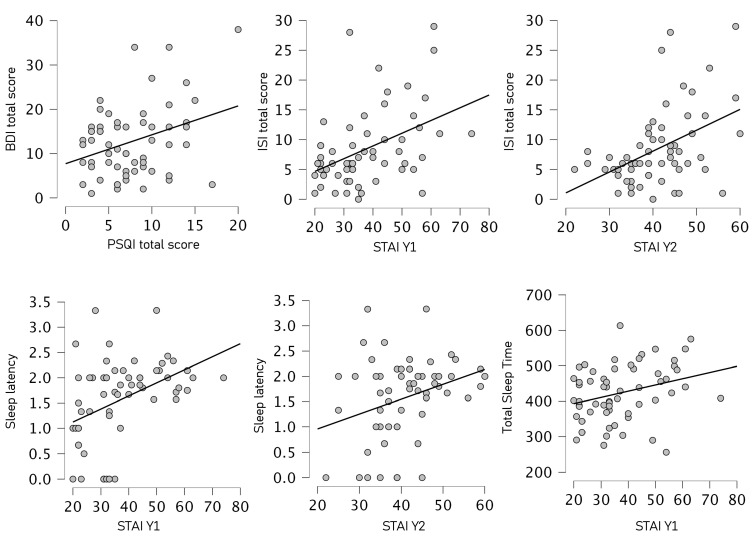
Correlation matrix plots between clinical variables and sleep index. Note: STAI Y1 = State-Trait Anxiety Inventory State, STAI X2 = State-Trait Anxiety Inventory Trait, BDI-2 = Beck Depression Inventory II Edition, PSQI = Pittsburgh Sleep Quality Index.

**Figure 2 brainsci-13-00132-f002:**
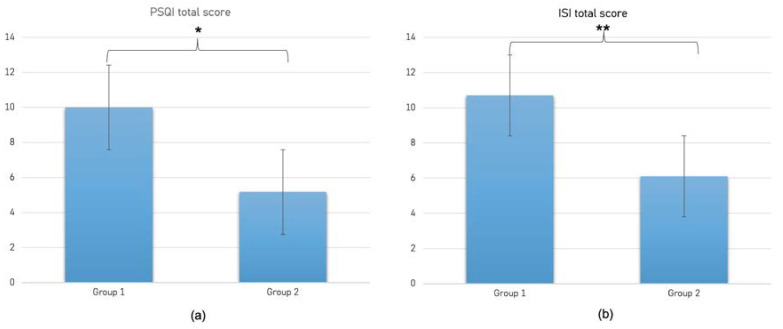
Comparison between Group 1 (prisoners) and Group 2 (controls) with respect to sleep quality as measured by PSQI (panel (**a**)) and insomnia complaints as measured by ISI (panel (**b**)); * *p* < 0.001; ** *p* = 0.005.

**Figure 3 brainsci-13-00132-f003:**
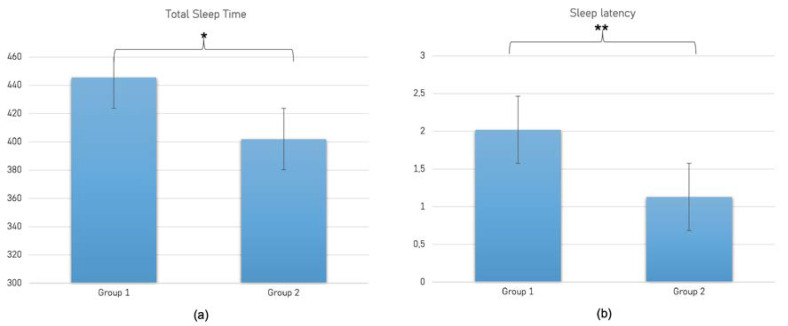
Comparison between Group 1 (prisoners) and Group 2 (controls) with respect to Total Sleep Time (panel (**a**)) and Sleep Latency (panel (**b**)) as measured by means of actigraphy; * *p* = 0.031; ** *p* < 0.001.

**Table 1 brainsci-13-00132-t001:** Pearson’s r (and related level of significance) between BDI-2, STAI, PSQI, ISI, and main actigraphic variables.

	BDI-AS		BDI-C		BDI Total Score		STAI Y1		STAI Y2		PSQI Tot		ISI Severity		ISI Impact		ISI Satisfaction		ISI Total Score		Latency		Efficiency		TST		WASO	
	*r*	*p*	*r*	*p*	*r*	*p*	*r*	*p*	*r*	*p*	*r*	*p*	*r*	*p*	*r*	*p*	*r*	*p*	*r*	*p*	*r*	*p*	*r*	*p*	*r*	*p*	*r*	*p*
**BDI-AS**																												
**BDI-C**	0.58	**<0.001**																										
**BDI total score**	0.933	**<0.001**	0.82	**<0.001**																								
**STAI Y1**	0.518	**<0.001**	0.213	0.102	0.462	**<0.001**																						
**STAI Y2**	0.464	**<0.001**	0.228	0.079	0.44	**<0.001**	0.718	**<0.001**																				
**PSQI tot**	0.385	**0.002**	0.151	0.251	0.331	**0.01**	0.607	**<0.001**	0.62	**<0.001**																		
**ISI severity**	0.102	0.438	0.193	0.14	0.188	0.15	0.385	**0.002**	0.424	**0.001**	0.499	**<0.001**																
**ISI impact**	0.168	0.201	0.166	0.204	0.207	0.113	0.31	**0.016**	0.311	**0.015**	0.455	**<0.001**	0.65	**<0.001**														
**ISI satisfaction**	0.258	**0.047**	0.224	0.085	0.299	**0.02**	0.475	**<0.001**	0.471	**<0.001**	0.479	**<0.001**	0.732	**<0.001**	0.729	**<0.001**												
**ISI total score**	0.193	0.14	0.218	0.095	0.257	0.048	0.437	**<0.001**	0.452	**<0.001**	0.535	**<0.001**	0.9	**<0.001**	0.872	**<0.001**	0.914	**<0.001**										
**Latency**	0.195	0.135	0.004	0.977	0.137	0.298	0.413	**0.001**	0.297	**0.021**	0.265	0.041	0.238	0.068	0.017	0.9	0.253	0.051	0.197	0.132								
**Efficiency**	0.128	0.33	0.271	0.037	0.2	0.126	0.089	0.501	−0.01	0.942	0.045	0.734	0.084	0.524	0.079	0.548	0.07	0.595	0.087	0.509	−0.211	0.106						
**TST**	0.1	0.447	0.264	**0.042**	0.201	0.124	0.295	**0.022**	0.156	0.233	0.127	0.333	0.142	0.278	0.017	0.896	0.252	0.052	0.157	0.232	0.058	0.66	0.311	**0.015**				
**WASO**	−0.119	0.363	−0.104	0.431	−0.111	0.398	0.022	0.866	0.057	0.666	−0.024	0.853	0.007	0.96	−0.112	0.393	0.027	0.84	−0.025	0.848	0.182	0.164	−0.84	**0.001**	0.204	0.118		

Note: STAI Y1 = State-Trait Anxiety Inventory State, STAI X2 = State-Trait Anxiety Inventory Trait, BDI-2 = Beck Depression Inventory II Edition, PSQI = Pittsburgh Sleep Quality Index, ISI = Insomnia, Severity Index, TST = Total Sleep Time, WASO = Wake After Sleep Onset. Significant correlations are indicated in bold.

## Data Availability

The data presented in this study are available upon request from the corresponding author. The data are not publicly available due to the privacy of the participants and rules regulating the participants’ life.

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
