# Peer review of "Reduced Sleep Amount and Increased Sleep Latency in Prisoners: A Pilot Study in an Italian Jail"

_brainsci, 2023, doi:10.3390/brainsci13010132_

Round 1

Reviewer 1 Report

The paper of D’Aurizio et al. investigates a very relevant topic, related to health in Jail. Specifically, the authors investigated the incidence of insomnia troubles in prisoners living in jail. Main results indicated a worsened mood in prisoners than in controls (with increased anxiety and depression) as well as a subjectively reported low sleep quality with a clinical presence of insomnia complaints. work is well. conducted and well described, and the limits are well highlighted in the conclusions (especially for the less representative sample). I suggest modifying the images by inserting, where necessary, the significance asterisk and inserting the reference to the significant results in the captions so that it is more readable. It would also be useful to insert a graph with the most significant correlations.

Author Response

REVIEWER 1

The paper of D’Aurizio et al. investigates a very relevant topic, related to health in Jail. Specifically, the authors investigated the incidence of insomnia troubles in prisoners living in jail. Main results indicated a worsened mood in prisoners than in controls (with increased anxiety and depression) as well as a subjectively reported low sleep quality with a clinical presence of insomnia complaints. work is well. conducted and well described, and the limits are well highlighted in the conclusions (especially for the less representative sample).

Authors’ Answer (AA): We thank the Reviewer for his/her interest and positive impression on our manuscript.

I suggest modifying the images by inserting, where necessary, the significance asterisk and inserting the reference to the significant results in the captions so that it is more readable. It would also be useful to insert a graph with the most significant correlations.

AA: We thank the Reviewer for his/her suggestions: the figures have been changed accordingly and the correlation graphs have been inserted.

Reviewer 2 Report

The manuscript “An actigraphic sleep study in prisoners: a pilot study in an Italian jail” proposed an investigation aimed to assess quality and quantity of sleep by using for the first time actigraphy on a sample of 30 male prisoners and 30 age-matched control volunteers. Along with variables measured by actigraphy for seven consecutive nights, several other mood- and subjective sleep-related measures were taken, i.e. state and trait anxiety, depression, sleep quality, and insomnia severity. Results showed prisoners had a significant lower mood than control participants as well as a worse self-reported sleep quality. Those results were confirmed by actigraphy data, showing a longer sleep onset and a reduced total sleep time in prisoners with respect to control participants. Authors discussed their results in light of previous literature addressing the limitations of their study as well as giving hints for further research.

I carefully read the manuscript, and I think it may be of interest for the readers of Brain sciences. Nevertheless, I think that could be worth considering some minor points before publication. Below there are my comments and suggestions.

Introduction section

The manuscript is very well-written and properly addresses the interesting issue of the sleep quality and quantity in a special population, i.e., jail prisoners. Although this is a pilot study, its relevance is given by the fact that sleep-related variables are assessed in a both objective and subjective way, paving the road to larger cross-sectional and longitudinal studies. The introduction section as well as the aims of the study are clear and detailed.

Materials and Methods section

2.2. Materials subsection: Please include a reliability estimate (e.g., Cronbach’s alpha) for each paper-and-pencil test.

Table 1: Please delete the upper off-diagonal triangle of correlations since it’s specular to the lower one. Also, the diagonal of 1’s could be deleted.

Results section

Results are clear and detailed. I only would try to improve the quality of the figures. I also suggest for the next studies to perform multivariate analysis for the dimensions of the same test. I understand that this time such analysis is not feasible due to low sample size.

Discussion section

The explanations provided in the discussion section are accurate and supported by previous literature. I also appreciated the mention of limitations as well as of future research directions.

Author Response

REVIEWER 2

The manuscript “An actigraphic sleep study in prisoners: a pilot study in an Italian jail” proposed an investigation aimed to assess quality and quantity of sleep by using for the first time actigraphy on a sample of 30 male prisoners and 30 age-matched control volunteers. Along with variables measured by actigraphy for seven consecutive nights, several other mood- and subjective sleep-related measures were taken, i.e. state and trait anxiety, depression, sleep quality, and insomnia severity. Results showed prisoners had a significant lower mood than control participants as well as a worse self-reported sleep quality. Those results were confirmed by actigraphy data, showing a longer sleep onset and a reduced total sleep time in prisoners with respect to control participants. Authors discussed their results in light of previous literature addressing the limitations of their study as well as giving hints for further research.

I carefully read the manuscript, and I think it may be of interest for the readers of Brain sciences. Nevertheless, I think that could be worth considering some minor points before publication. Below there are my comments and suggestions.

AA: We thank the Reviewer for his/her interest and positive impression on our manuscript.

Introduction section

The manuscript is very well-written and properly addresses the interesting issue of the sleep quality and quantity in a special population, i.e., jail prisoners. Although this is a pilot study, its relevance is given by the fact that sleep-related variables are assessed in a both objective and subjective way, paving the road to larger cross-sectional and longitudinal studies. The introduction section as well as the aims of the study are clear and detailed.

AA: We thank again the Reviewer for his/her interest and positive impression on our manuscript.

Materials and Methods section

2.2. Materials subsection: Please include a reliability estimate (e.g., Cronbach’s alpha) for each paper-and-pencil test.

AA: In the current version of the manuscript, Cronbach’s alpha reliability coefficients have been included for each questionnaire.

Table 1: Please delete the upper off-diagonal triangle of correlations since it’s specular to the lower one. Also, the diagonal of 1’s could be deleted.

AA: We thank the Reviewer for his/her suggestion: the correlations table has been changed accordingly.

Results section

Results are clear and detailed. I only would try to improve the quality of the figures. I also suggest for the next studies to perform multivariate analysis for the dimensions of the same test. I understand that this time such analysis is not feasible due to low sample size. 

AA: We thank the Reviewer for his/her concern: the figures have been changed accordingly. We also thank for the interesting suggestion.

Discussion section 

The explanations provided in the discussion section are accurate and supported by previous literature. I also appreciated the mention of limitations as well as of future research directions.

AA: We thank the Reviewer for his/her positive impression on our manuscript.

Reviewer 3 Report

this is a very nice study and novel some comments to improve the paper.

 the title is not informative and doesn't reflect the content of the study.

the abstract is very brief and doesn't summarize the study.

the introduction failed to address the importance of this study the fact that actigraphy or wearable devices were not being used in the jail it's not a very good excuse. authors need to show the importance of this study to the field of sleep medicine and how can this noninvasive cheap and relatively safe device of actigraphy can be an important assessment tool for designing plans regarding to improvement of sleep.

describe your precise goals and any established hypothesis.

early in the text, highlight the important study design components. this should be a cross-sectional study. kindly not that this is very important for also discussing the limitation of this study.

describe the context, the places, the pertinent times, such as the recruiting, exposure, follow-up, and data collection times. when was this study performed. pls more details.

give the qualifying requirements (did u exclude people with certain issues e.g. substance use or thyroid), as well as the sources and procedures used to choose the participants.

all outcomes, exposures, predictors, potential confounders, and effect modifiers should be precisely defined.

describe how the study size was determined. this is very important to determine if this is study by any chance is dependable or is only a pilot. i opt for pilot.

describe the methods used to handle quantitative variables in the analyses. describe the classifications that were chosen, if relevant, and why.

describe all statistical techniques, including confounding correction techniques. i am note sure why used anova when t-test was better and also why not reporting effect size explicitly.

describe any techniques used to study interactions and subgroups. age- time in prison- and others..

describe how missing data were handled. i guess some missing data was present and its unlikely to get complete for long 4-5 scales.

summarize the main findings in relation to the study's goals.

describe the study's limitations while considering any possible bias or imprecision sources. discuss any potential bias's magnitude and direction.

give a cautious overall interpretation of the findings while taking into account the goals, restrictions, variety of analyses, outcomes from related studies, and other pertinent data.

discuss how broadly (externally valid) the study's findings can be applied. again this is pilot and also cross-sectional. these are two main limitations.

Author Response

REVIEWER 3

this is a very nice study and novel some comments to improve the paper.

AA: We thank the Reviewer for his/her positive impression on our manuscript.

 the title is not informative and doesn't reflect the content of the study. 

AA: We changed the title to make it more informative on the content of the study.

the abstract is very brief and doesn't summarize the study. 

AA: We slightly changed the abstract by trying to increase its content. The length was kept reduced to comply with journal’ guidelines.

the introduction failed to address the importance of this study the fact that actigraphy or wearable devices were not being used in the jail it's not a very good excuse. authors need to show the importance of this study to the field of sleep medicine and how can this noninvasive cheap and relatively safe device of actigraphy can be an important assessment tool for designing plans regarding to improvement of sleep. 

AA: In the revised version of the manuscript, we included some hints on actigraphy importance.

describe your precise goals and any established hypothesis.

AA: In the revised version of the manuscript, we included goals and hypotheses.

early in the text, highlight the important study design components. this should be a cross-sectional study. kindly not that this is very important for also discussing the limitation of this study. 

AA: In the revised version of the manuscript, we highlighted the nature of the study.

describe the context, the places, the pertinent times, such as the recruiting, exposure, follow-up, and data collection times. when was this study performed. pls more details. 

AA: In the revised version of the manuscript, we integrated some info with the ones already present in the previous version.

give the qualifying requirements (did u exclude people with certain issues e.g. substance use or thyroid), as well as the sources and procedures used to choose the participants .all outcomes, exposures, predictors, potential confounders, and effect modifiers should be precisely defined. 

AA: We thank the Reviewer for these concerns: in the revised version of the manuscript, we integrated some info with the ones already present in the previous version.

describe how the study size was determined. this is very important to determine if this is study by any chance is dependable or is only a pilot. i opt for pilot. 

AA:  We thank the Reviewer for the comment: due to the particular environment, the study size was determined by voluntary adhesion, thus the generalizability is surely limited. We agree with the Reviewer about the kind of study: how described both in the title and the main text this study has to be seen as a pilot study.

describe the methods used to handle quantitative variables in the analyses. describe the classifications that were chosen, if relevant, and why. 

AA:  All points were already present in the previous version of the manuscript.

describe all statistical techniques, including confounding correction techniques. i am note sure why used anova when t-test was better and also why not reporting effect size explicitly.

AA: We thank the Reviewer for the comment: Considering the sample size, we thought it more correct to perform the one-way ANOVA than the t-test. With respect to effect size, for each significant comparison, we have now reported ηp2.

describe any techniques used to study interactions and subgroups. age- time in prison- and others..

AA: We thank the Reviewer for the comment: as previous study we tried to take into consideration age and time-in-prison. But since age difference between groups was not significant, we did not include it in the analysis. With respect to time-in-prison again it was quite constant through the group (12,1±2,4 ys) and there was no reason to include it as a possible covariate.

describe how missing data were handled. i guess some missing data was present and its unlikely to get complete for long 4-5 scales. 

AA: We thank the Reviewer for the comment: we had no missing data. Our participants were very motivated and filled in all scales and questionnaires.

summarize the main findings in relation to the study's goals

AA: In the revised version of the manuscript, in the first paragraphs of the results findings follow goals and hypotheses as listed in the final para of the introduction.

describe the study's limitations while considering any possible bias or imprecision sources. discuss any potential bias's magnitude and direction.

AA: In the revised version of the manuscript, limitations and possible biases are already included and discussed.

give a cautious overall interpretation of the findings while taking into account the goals, restrictions, variety of analyses, outcomes from related studies, and other pertinent data.

AA: As said before all these points have been addressed in the limitations paragraph.

discuss how broadly (externally valid) the study's findings can be applied. again this is pilot and also cross-sectional. these are two main limitations

AA: All these points have been addressed in the limitations paragraph.

Round 2

Reviewer 3 Report

thank you for addressing my concerns.